# Fast and Memory-Efficient Video Diffusion Using Streamlined Inference

**Zheng Zhan**[1][*]  **Yushu Wu**[1][*]  **Yifan Gong**[1]  **Zichong Meng**[1]  **Zhenglun Kong**[12]
**Changdi Yang**[1]  **Geng Yuan**[3]  **Pu Zhao**[1][†]  **Wei Niu**[3]  **Yanzhi Wang**[1]
[1]Northeastern University  [2]Harvard University  [3]University of Georgia

512×768 resolution

Animatediff
Peak Mem: 29.7G
Latency: 16.9s

Ours
Peak Mem:
9.08G **(3.3× less)**
Latency:
10.7s **(1.6× less)**

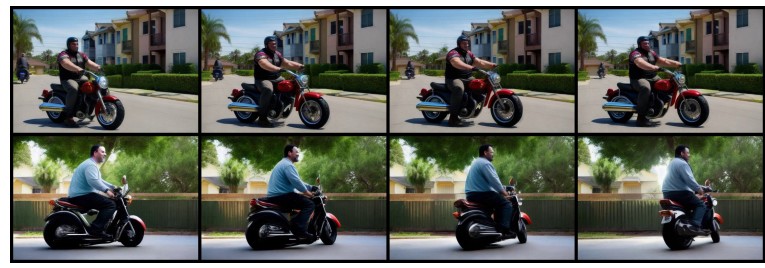

a full-sized man rides a comically small motorcycle through a residential neighborhood

Figure 1: Our Streamlined Inference is a training-free inference framework for video diffusion models that can reduce the computation and peak memory cost without sacrificing the quality.

## Abstract

The rapid progress in artificial intelligence-generated content (AIGC), especially with diffusion models, has significantly advanced development of high-quality video generation. However, current video diffusion models exhibit demanding computational requirements and high peak memory usage, especially for generating longer and higher-resolution videos. These limitations greatly hinder the practical application of video diffusion models on standard hardware platforms. To tackle this issue, we present a novel, training-free framework named Streamlined Inference, which leverages the temporal and spatial properties of video diffusion models. Our approach integrates three core components: Feature Slicer, Operator Grouping, and Step Rehash. Specifically, Feature Slicer effectively partitions input features into sub-features and Operator Grouping processes each sub-feature with a group of consecutive operators, resulting in significant memory reduction without sacrificing the quality or speed. Step Rehash further exploits the similarity between adjacent steps in diffusion, and accelerates inference through skipping unnecessary steps. Extensive experiments demonstrate that our approach significantly reduces peak memory and computational overhead, making it feasible to generate high-quality videos on a single consumer GPU (e.g., reducing peak memory of AnimateDiff from 42GB to 11GB, featuring faster inference on 2080Ti)[1].

## 1 Introduction

Recent years have witnessed continual progress and advancements in artificial intelligence-generated content (AIGC). Among them, diffusion models allow artists and amateurs to create visual content

---

[*]Equal contributions.

[†]Corresponding author

[1]Code available at: `https://github.com/wuyushuwys/FMEDiffusion`

38th Conference on Neural Information Processing Systems (NeurIPS 2024).

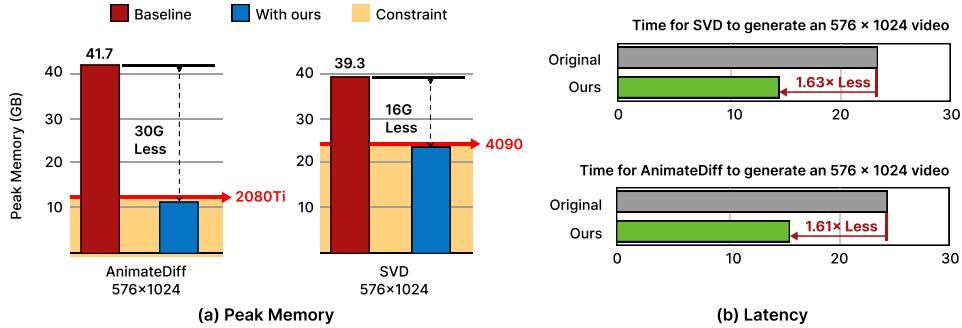

Figure 2: Comparison on Animatediff and SVD inference using our Streamlined Inference. Memory requirement is crucial as **"Out of Memory"** errors prevent the GPU from performing inference.

with text prompts, advancing the development of image and video generation in both academia and industry. For video diffusion models, the latest works such as SVD-XT [3], Gen2 [29], Pika [20], and notably the more advanced Sora [28], demonstrate impressive capabilities in producing visually striking and artistically effective videos. Despite their great performance, video diffusion models also exhibit high computational requirements and substantial peak memory, particularly when generating longer videos with higher resolutions. For instance, SVD-XT generates 25 frames simultaneously with a resolution of $576 \times 1024$, while Sora expands these capabilities by supporting the generation of longer videos (over a minute) at a higher resolution of $1080 \times 1920$. Given the trends of generating longer videos with higher quality, the escalating memory and computation demands have impeded practical applications of these large-scale video diffusion models on various platforms.

Existing model compression methods to reduce peak memory and latency, such as weight pruning [38, 12, 35, 6, 48, 23, 42, 44, 47], quantization [37, 22, 32], and distillation [18, 10, 43, 39], typically require substantial retraining or fine-tuning of the compressed model to recover performance. This process is costly, time-consuming, and may raise data privacy concerns. Applying these methods in zero-shot avoids the expensive retraining, but leads to severe performance degradation. Furthermore, the variety and complexity of video diffusion architectures further complicate the model optimization. Therefore, it is challenging yet crucial to develop an effective and efficient video diffusion framework with reduced computations, smaller peak memory and less data (no re-training) requirements for its wide applications.

To address the above challenges, we first identify the sources of the **high computation and memory cost**, which scale up with the iterative denoising process and the simultaneous processing of multiple frames. We further observe that the feature maps of certain layers may exhibit high similarity between multiple consecutive denoising steps due to the temporal property of videos, enabling further optimizations for acceleration. Based on that, we propose a training-free framework named Streamlined Inference, by leveraging the temporal and spatial characteristics of video diffusion models to effectively reduce peak memory and computational demands. Our framework contains three core components: Feature Slicer, Operator Grouping, and Step Rehash, which work together closely and comprehensively with different focuses on peak memory reduction or inference acceleration.

Our Feature Slicer performs lossless feature slicing in both temporal and spatial layers, raising the possibility of peak memory reduction through processing smaller features. However, the feature slicer alone is not able to decrease peak memory as we still need to store all intermediate results of one layer for all sliced features to form a complete intermediate feature map for the next layer. To reduce peak memory practically, we further propose Operator Grouping to group homogeneous and consecutive operators in the computational graph. Within each operator group, the intermediate result of one sliced feature can be directly sent to the next operator/layer without waiting for aggregation with all other intermediate results, achieving the full potential of Feature Slicer to reduce the peak memory. Furthermore, a pipeline technique is proposed to accelerate the computations of the same operator group for multiple sliced feature inputs, with improved parallelism.

Moreover, observing the high similarity of certain features between multiple consecutive denoising steps, we propose Step Rehash to reuse the generated features for a few following steps due to their high similarity, skipping the exact expansive and repetitive generation of similar features and thereby accelerating the video diffusion significantly. With this framework, we can generate high-quality

videos in a fast and memory-efficient manner on a single consumer GPU, as shown in Fig. 2. For example, the peak memory of AnimateDiff [11] can be reduced significantly from 41.7GB to 11GB, featuring inference on a typical consumer GPU 2080Ti. We summarize our contributions as follows:

- We propose a novel training-free framework that can significantly reduce the peak memory and computation cost for the inference of video diffusion models by leveraging the spatial and temporal characteristics of video diffusion models.

- Our approach can be seamlessly integrated into existing video diffusion models. Our extensive experiments on SVD, SVD-XT, and AnimateDiff demonstrate our effectiveness to reduce peak memory and accelerate inference without sacrificing quality.

- Our approach offers a new research perspective for fast and memory-efficient video diffusion, enabling the generation of higher quality and longer videos on consumer-grade GPUs.

## 2   Related Work

**Video Diffuison Models.** For video generation, various approaches have been proposed, with VDM [17] as a leading example. VDM transforms the conventional U-Net [30] architecture of image diffusion models into a 3D U-Net structure, employing joint training on both images and videos. MagicVideo [49] is the first work that introduces Latent Diffusion Model (LDM) for text-to-video (T2V) generation in latent space. LVDM [13] introduces a mask sampling technique that enhances its longer video generation capability. ModelScope [36] incorporates spatial-temporal convolution and attention into LDM. Video LDM [4] trains a T2V network composed of three training stages, enabling higher quality and longer video generation. Show-1 [45] first introduces the fusion of pixel-based and latent-based diffusion models for T2V generation. Recently, Stable Video Diffusion (SVD) [3] identifies three key stages for training video LDMs: text-to-image (T2I) pretraining, video pretraining, and high-quality video finetuning.

**Architectural Efficiency of Video Diffusion Models.** There are various research efforts exploring either architectural efficiency or model compression techniques for image/video generation. For example, ED-T2V [24] freezes parameters to reduce training costs and proposes a attention mechanism to ensure temporal coherence. SimDA [40] devises a parameter-efficient training approach by maintaining the parameter of the T2I model and uses two adapters to train it. For model compression, Diff-pruning [6] employs structural pruning techniques to reduce inference time at each sampling step. Additionally, the work [22] implements quantization on diffusion models using low-precision data types. However, these methods take substantial efforts to retrain or finetune the diffusion model to recover performance, which is costly, time-consuming, and may raise data privacy concerns. Furthermore, applying post-training compression techniques in one-shot [8, 7, 34] may save the retraining/fine-tuning efforts, but suffers from significant performance degradation.

**Sample-Efficient Video Diffusion Models.** To address the iterative denoising process in diffusion models and improve the sampling efficiency, two approaches are proposed. The first approach [2, 19, 5, 25] focuses on creating rapid solvers to resolve the differential equation associated with the denoising process more effectively. The works [31, 27, 21] utilize knowledge distillation methods to compress and simplify the sampling trajectory efficiently, thereby enhancing overall performance. Imagen video [15] is one of the first methods to apply progressive distillation on video diffusion models, incorporating guidance and stochastic samplers. Recent work Deepcache [26] proposes a novel training-free paradigm that accelerates diffusion models by reusing the high-level features.

## 3   Motivation

**Peak memory and computation analysis.** Existing open-source video diffusion models [11, 3, 36, 46] typically adopt a pretrained T2I 2D-UNet as backbone. Their temporal layers are seamlessly integrated into the backbone 2D-UNet, positioned after every spatial layer. Here, we use SVD as an example to demonstrate how peak memory and computational overhead scale with the number of frames. The SVD model is trained with two distinct configurations: regular SVD is designed to generate 14 frames, while SVD-XT is tailored to produce 25 frames. To generate 14 or 25 video frames concurrently with SVD, its latent features require massive GPU memory and computation consumption, estimated to be approximately $14\times$ or $25\times$ higher than its base T2I model. This

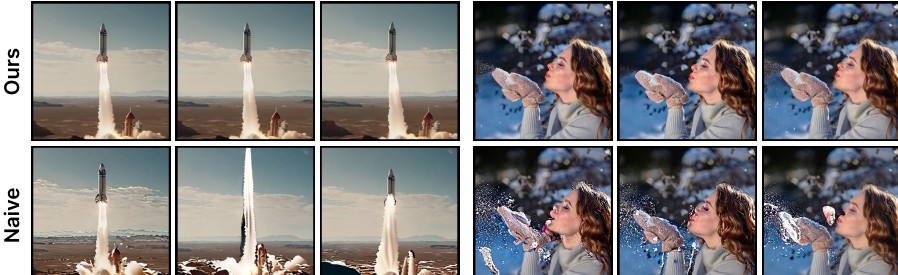

Figure 3: The quality results of our method and naïve slicing. Note that naïve slicing will incur unpleasant artifacts due to lack of temporal correction by fewer frames.

estimate does not even account for the additional memory required by SVD's extra-temporal layers. More specifically, the SVD consumes 39.49G of peak memory for $576 \times 1024$ resolution output, whereas its image generation counterpart only requires 6.33G of memory at the same resolution. Furthermore, incorporating the classifier-free guidance [16] substantially enhances the generation quality but doubles the peak memory required during inference.

Consequently, video diffusion is computationally demanding, but the challenge of memory consumption is more critical and demands immediate attention. Most consumer-grade GPUs do not have enough memory for video diffusion models and, therefore, suffer from the **"Out of Memory"** error, which prevents the GPU from generating high-quality videos. There is no workaround without switching GPUs. Most users have to endure generating short and low-resolution videos.

**Naïve Slicing.** A Naïve approach to reduce peak memory is to execute the video diffusion inference clip-by-clip. However, this strategy is hindered by the temporal layers, which are essential for maintaining temporal correlation in video diffusion models. Forcibly implementing this approach can generate random artifacts and cause motion vanishing in the output video, as detailed in Fig. 3. Therefore, designing a memory-efficient inference framework is a challenging and non-trivial task.

## 4 Streamlined Inference Framework

To address the above massive peak memory and computation costs, in this section, we propose a training-free framework named Streamlined Inference, which is composed of three core components: Feature Slicer, Operator Grouping, and Step Rehash. First, we discuss Feature Slicer, designed to partition input features of spatial and temporal layers, and enable the potential of massive peak memory reduction. Next, we introduce our Operator Grouping technique to aggregate homogeneous and consecutive operators into the same group, achieving the full potential of Feature Slicer to reduce peak memory through reusing the memory of intermediate result from previous sliced feature. Finally, we discuss our Step Rehash method to reuse the same feature for a few consecutive steps due to their high similarity. It accelerates the inference without increasing peak memory overhead as it skips certain denoising steps with less computations.

### 4.1 Feature Slicer

Video diffusion models contain spatial and temporal layers which extract the corresponding information from their specific domains. On this basis, we propose a feature slicer that consists of two components: Spatial-layer slicer and Temporal-layer slicer, to divide the feature map into multiple batches/sub-features, ensuring accurate computation without introducing additional operations. The slicer is further utilized for Operator Grouping to reduce peak memory cost.

**Spatial layers slicer.** Based on our profiling (more details can be found in Appendix A) for memory allocation of various video diffusion models, we find that performing slices at spatial layers can greatly reduce the memory footprint. The 5-D feature in the spatial layer $X \in \mathbb{R}^{B \times T \times C \times H \times W}$ can be reshaped to a 4-D feature $X \in \mathbb{R}^{(B \times T) \times C \times H \times W}$, where $B, T, C, H, W$ are the batch size, number of frames, channels, height, and width, respectively. Thus, we slice it into $k$ sub-features, $\left\{ X_i \in \mathbb{R}^{n_i \times C \times H \times W} \right\}_{i=1}^{k}$, where $n_i = \lceil B \times T / k \rceil$ with $\lceil \cdot \rceil$ denoting the least integer greater than or equal to the input. If $\lceil B \times T / k \rceil \neq B \times T / k$, the dimension of the last sub-feature $n_k$ is different

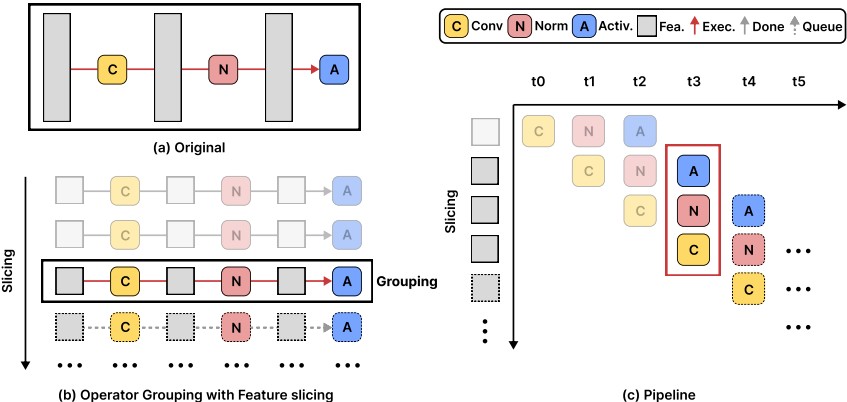

Figure 4: Overview of Operator Grouping with Pipeline in our framework.

from others. The spatial layer slicer is applicable for most operations in spatial layers such as `Conv2D`, `GroupNorm`, `LayerNorm`, `Attention`, and `Linear`.

**Temporal layers slicer.**  The input of the temporal layer is a 5-D feature map with dimensions {*batch, channels, frames, height, width*}. 3-D operations such as `Conv3D` are employed to extract temporal information from the 5-D feature. Differing from spatial layers, slicing along the temporal dimension may disrupt the extraction and processing of temporal information. Therefore, we keep the temporal dimension untouched while slicing over other dimensions. Specifically, the 5-D feature $X \in \mathbb{R}^{B \times T \times C \times H \times W}$ can be sliced to $k_h \times k_w$ sub-features $\left\{ X_{ij} \in \mathbb{R}^{B \times T \times C \times h_i \times w_j} \right\}_{i=1,j=1}^{i=k_h,j=k_w}$, where $h_i = \lceil H/k_h \rceil$ and $w_j = \lceil W/k_w \rceil$. After detailed profiling different configurations for temporal layer slicer, we discover that the configuration with $k_h = \max(H, 16)$ and $k_w = \max(W, 16)$ can result in promising peak memory reduction.

## 4.2   Operator Grouping

Although `Feature Slicer` converts the original feature map into multiple smaller sub-features with reduced memory footprint, the peak memory can not be reduced since the intermediate results of multiple sliced features require re-consolidation to send to the next layer/operator as inputs. It still needs to store all intermediate outputs of sliced features to form the united/unsliced intermediate feature map without actual peak memory reduction. Therefore, to address this problem, we propose `Operator Grouping` to group the operators accordingly in the computational graph, achieving the full potential of `Feature Slicer` with effective peak memory reduction due to less memory reserved for intermediate results. Furthermore, a pipeline technique is proposed to optimize the inference of operator groups with improved parallelism and practical acceleration.

### 4.2.1   Grouping Operators for Peak Memory Reduction

`Operator Grouping` directly re-uses existing operators by aggregating consecutive homogeneous operators into the same group. Homogeneous operators indicate these operators extract features from coherent domains and dimensions. In video diffusion models, different operators can be grouped into $\text{GroupOP}^t$ (temporal operator groups) and $\text{GroupOP}^s$ (spatial operators groups) to ensure the well-preserved semantics of sliced sub-features within each group. For example, in the SVD Model [3], consecutive `GroupNorm`, `Conv2D`, `SiLU`, and `Up/DownSample` operators in the *Spatial ResBlock* can be aggregated to one group, as these operators all extract features from spatial domain and are deemed homogeneous. As in Fig. 4, when computing the output feature $X^o$ for an operator group (GroupOP), the input feature $X$ is sliced into multiple sub-features $X_1, X_2, \ldots, X_k$ with Feature Slicer. Each sub-feature $X_i$ goes through the operator group and their outputs are concatenated after all outputs are available, as shown in Eq. (1),

$$X^o = \text{Concat}\left(\text{GroupOP}(X_1), \text{GroupOP}(X_2), \ldots, \text{GroupOP}(X_k)\right) \qquad (1)$$

where $k$ is the number of slices, and `Concat` is the concatenation operation.

**Reducing peak memory cost.**  As shown in Fig. 4, the peak memory with the operator group is determined by the memory footprint of the input feature, the output feature, and the intermediate

results. Without operator grouping, all intermediate results of all operators for sliced sub-features will allocate their own memory, hence failing to reduce memory consumption. Compared with the case above, grouped operators only need to allocate memory for intermediate results of a single sliced sub-feature and the final outputs, without the necessity to allocate full intermediate features corresponding to the original unsliced input feature, as shown in Fig. 4 (a) and (b). Operator Grouping can effectively reduce peak memory cost, enabling successful inference of video diffusion models on one single consumer or commercial GPU with low or moderate available memory, as shown in Tab. 1.

**Mitigating I/O intensity.** As the original feature map is sliced into multiple sub-features to reduce peak memory cost, the computation may slow down due to multiple iterations corresponding to multiple inputs. However, we surprisingly observe that even with the naive basic for-loop implementation for each sub-feature as shown in Fig. 4 (b), the overall runtime with Operator Grouping is around 10% slower than that of the original unsliced version. The marginal increase in runtime can be attributed to the memory bound of the GPU for video diffusion inference. Specifically, current video diffusion model inference suffers from the memory bound, where the I/O overhead of intermediate results is more notorious than their computation workload. The slicer provides a solution to mitigate the I/O burden, thus balancing the computation and memory read/write to fully utilize the GPU capacity.

### 4.2.2 Pipelining with Improved Parallelism and Practical Acceleration.

With the proposed `Feature Slicer` and `Operator Grouping`, the peak memory will decrease significantly with a marginal increase for the computation runtime (based on the basic for-loop implementation). With a deeper investigation for the computation patterns, we find that the *for-loop implementation* cannot maximize GPU parallelism, and further employ the pipelining technique to optimize the *for-loop implementation* for faster inference without additional memory cost.

With `Operator Grouping`, there are multiple operators in one group to process one sliced sub-feature sequentially. With the naive *for-loop implementation*, before feeding each sliced sub-feature into another operator group, it needs to wait until the last sub-feature finishes its computation within the group. The parallelism can be further improved with the proposed pipeline method. Specifically, in an operator group, after a sliced feature map is computed by the out-of-place computation operator (e.g., `Conv`, `GroupNorm`, `Attention`, etc.) and sent to the next operator, its previous allocated memory is no longer required, but it is still reserved during inference, leading to resource waste. We can pipeline all operators in the same group to mitigate this issue. As shown in Fig. 4(c), once the `Conv` operator completes processing a sliced feature $X_i$ as described in Eq. (1) and its outputs are sent to the next operator `Norm`, the subsequent sliced feature $X_{i+1}$ is immediately piped into the same `Conv` operator, reusing the reserved memory of $X_i$. In this way, multiple operators are executed simultaneously with improved parallelism. No additional memory is required, as we only make use of previously reserved memory.

**Acceleration performance.** With the naive *for-loop implementation*, only one operator in an operator group is executed at a time. However, our pipeline method can simultaneously execute multiple operators (such as `Conv`, `Norm`, and `Activation` as depicted in Fig. 4 (c)) without incurring additional memory. Consequently, the inference speed can be further improved. Accordingly, integrating the pipeline within Operator Grouping can mitigate 10% speed degradation caused by feature slicing.

## 4.3 Step Rehash

In this section, we further introduce our step rehash method to optimize the iterative denoising steps for effective acceleration in video diffusion generation. Capitalized on the high similarity between adjacent steps, our approach accelerates the video generation, while ensuring both high quality and temporal consistency across video frames, without extra memory cost. Next we first discuss our observations for the high feature similarity and then explain details of our step rehash.

### 4.3.1 Similarity of Temporal Features between Steps

**Similarity visualization.** The denoising process of U-Net in diffusion models requires multiple steps and the features of different steps may share certain similarities with minor differences [26]. To explore this, we analyze the feature maps averaged over multiple images at different parts of the model and plot the similarity between features of different steps, with an example shown in Fig. 5 (and more results and details demonstrated in Appendix C). We find two key insights below:

- The similarity between adjacent steps significantly depends on certain blocks and layers, and it may change sharply after specific operations in video diffusion. The features do not always show high similarity. For example, neither deeper layers within the same block nor those in middle blocks consistently show higher similarity between adjacent steps.
- The features between adjacent steps following the temporal layers and spatial layers in video diffusion usually exhibit remarkably higher similarity compared to outputs of other layers.

**High similarity after temporal layers.** Existing video diffusion models typically employ pretrained image diffusion models as their backbone. While these image models are trained to produce a variety of images, the addition of temporal layers is designed to improve the temporal continuity of latent features. This enhancement significantly strengthens their correlation, thereby increasing similarity among the features.

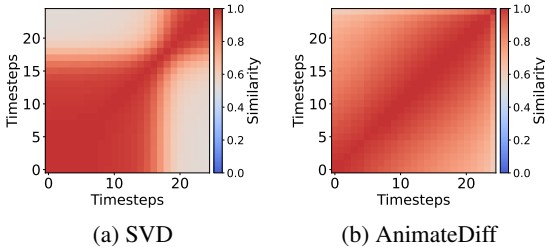

(a) SVD        (b) AnimateDiff

Figure 5: The high similarity of output features after temporal layers in $U_3$ between each timestep.

**Motivation and challenges for step rehash.** Due to the high similarity between features of different steps, we propose the step rehash method to skip the computation of certain features by reusing previous generated features. However, we need to address the challenges of when and where to skip. Specifically, based on the above insights, simply reusing features from deeper layers does not guarantee better results since deeper layers may not show high similarity. We need to carefully choose what layers or blocks can be skipped **(where to skip)** to make use of high similarities without significantly downgrading the generation performance. As shown in fig. 5, it exhibits high similarity between adjacent time steps, but the similarity pattern differs between video diffusion models. Thus, we need to determine which steps can use skip strategy **(when to skip)**, and the remaining steps that require full computation are *full computation steps*.

### 4.3.2 Step Rehash

Here we specify the details of our step rehash. The video diffusion models typically use a U-Net architecture with 4 down-sampling and 4 up-sampling blocks, and their output features can be represented by $D_{0\sim3}^s$ and $U_{0\sim3}^s$, respectively, with $s$ denoting the current step number as shown in Fig. 6. Typically, $U_b^s$ is obtained by feeding $D_b^s$ and $U_{b-1}^s$ into the $b^{th}$ ($b > 0$) up-sampling block, and $U_3^s$ is the final output of the $s^{th}$ step. Based on similarity analysis, in the next step $s + 1$, we can directly reuse the output features of the temporal layer from the previous step $s$ without actual exact computations. Our insights into the similarity indicate that deeper and middle blocks do not consistently demonstrate high similarity.

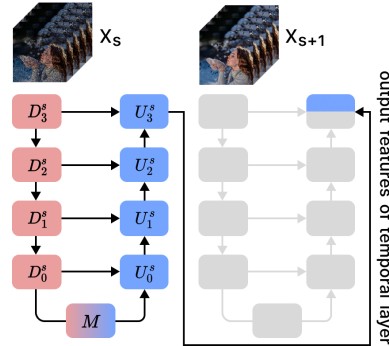

Figure 6: Illustration of Step Rehash. Computation in grey areas are skipped.

high similarity. Reusing their features results in significant degradation of generation quality. Therefore, we rehash features of the temporal layer in the final up-sampling block. Specifically, to obtain $U_3^{s+1}$ for step $s + 1$, we feed the output features of the temporal layer from $U_3^s$ (current *full computation step*) into the final up-sampling block. Note that we only compute part of $U_3^{s+1}$ and do not need to compute $D_3^{s+1}$ for concatenation, since our reused temporal layer is deeper than the `concat` operator for features from $D_3^{s+1}$, as shown in Fig. 6. We further propose a step search algorithm to solve the **when to skip** problem, algorithm details can be found in Appendix B.

## 5 Experimental results

### 5.1 Models, Datasets and Evaluation Metrics

We conduct the experiments on representative video diffusion models, including SVD [3], SVD-XT [3], and AnimateDiff [11]. For evaluation, we use the following evaluation protocols: The first frame of the video clips are extracted as the image condition for image-to-video generation and their captions are considered as the prompts. All experiments are conducted on a NVIDIA A100 GPU.

Table 1: Comparison of our Streamlined Inference with baseline methods in video visual quality (on UCF101), PM (Peak Memory), and latency (measured with 50 runs with the average value).

| Model | Method | FVD↓ | CLIP-Score↑ | $512 \times 512$ | | $576 \times 1024$ | |
|---|---|---|---|---|---|---|---|
| | | | | PM | Latency | PM | Latency |
| SVD #F=14 | Original | 307.7 | 29.25 | 20.91G | 10.23s | 39.49G | 23.29s |
| | Naïve Slicing | 1127.5 | 26.32 | 8.12G | 31.85s | 10.72G | 65.56s |
| | **Ours** | 340.6 | 28.98 | 13.67G | 7.36s | 23.42G | 14.24s |
| SVD-XT #F=25 | Original | 387.9 | 28.18 | 31.97G | 17.05s | 61.17G | 40.77s |
| | Naïve Slicing | 2180.0 | 24.42 | 8.12G | 59.86s | 10.72G | 121.82s |
| | **Ours** | 424.7 | 27.94 | 19.37G | 12.10s | 36.32G | 25.47s |
| AnimateDiff #F=16 | Original | 758.7 | 28.89 | 21.83G | 9.65s | 41.71G | 24.38s |
| | Naïve Slicing | 2403.9 | 26.63 | 7.22G | 19.98s | 9.92G | 38.69s |
| | **Ours** | 784.5 | 28.71 | 7.51G | 7.08s | 11.07G | 15.15s |

**Zero-shot UCF-101 [33]:** We sample clips from each categories of UCF-101 dataset, and gather a subset with 1,000 video clips for evaluation. Their action categories are considered as their captions. For SVD and SVD-XT, our samples are generated at a resolution of $576 \times 1024$ (14 frames for SVD and 25 frames for SVD-XT) and then resize to $240 \times 320$. For AnimateDiff, we generate samples with resolution $512 \times 512$ (16 frames).

**Zero-shot MSR-VTT [41]:** We generated a video sample for each of the 9,940 development prompts. The samples are at resolution $320 \times 576$ then resized to $240 \times 426$ for all models with different number of generated frames.

**Metrics:** We compute the FVD as outlined in [9] and CLIP-Score [14] using TorchMetrics [1] to measure the performance of generated samples.

**Baseline:** We use pretained weight for SVD (I2V) and AnimateDiff (T2V). We compare the proposed Streamlined Inference (use 13 *full computation steps*) with the original inference (use 25 *full computation steps*) and naïve slicing inference as mentioned in Sec.3. More specifically, for image-conditioned SVD model, we set each naïve slice with a frame size of 2 and use the last frames of each generated slice as the image condition for the next slice. For AnimateDiff, we evenly generate 4 slices with a frame size of 4, then combine them into a full video clip.

## 5.2 Quantitative Evaluation

The results from Table 1 demonstrate the effectiveness of our proposed method in managing memory, computational resources, and performance. Our method significantly reduced peak memory and latency while maintaining competitive FVD and CLIP-Score values across all three models and resolutions compared to the original method. For SVD, our method achieved a notable reduction in peak memory and latency while maintaining competitive FVD and CLIP-Score, unlike Naïve Slicing, which increased FVD and latency. For SVD-XT, our method improved over Naïve Slicing and balanced resource usage and performance. For AnimateDiff, our method significantly outperformed Naïve Slicing in FVD and latency, achieving nearly the same performance as the original method but with smaller latency and around a 70% reduction in peak memory.

## 5.3 Ablation Study

Our ablation study in Table 2 demonstrates that our Step Rehash method consistently outperforms DeepCache with the same number of *full computation steps*. Step Rehash skips more computations than DeepCache. For the SVD model, our method maintains competitive CLIP-Scores while slightly increasing FVD compared to the original method (FVD of 307.7 and CLIP-Score of 29.25 on UCF101). DeepCache performs poorly, increasing FVD and reducing video quality. For the AnimateDiff model, our method maintains stable FVD (603.9 vs. 607.13) and CLIP-Score (29.29 vs. 29.40) on MSR-VTT compared to the original method. DeepCache shows the worst performance on UCF101, with higher FVD and lower CLIP-Scores. Visual comparisons of our method with DeepCache are provided in Appendix D.

Table 2: Ablation study of our proposed method compared with DeepCache in video visual quality. Both our Step Rehash and DeepCache involve 13 *full computation steps*.

| Model | Method | UCF101 | | MSR-VTT | |
|---|---|---|---|---|---|
| | | FVD↓ | CLIP-Score↑ | FVD↓ | CLIP-Score↑ |
| SVD | Original | 307.7 | 29.25 | 373.6 | 26.06 |
| | DeepCache | 394.0 | 28.57 | 463.6 | 25.30 |
| | **Step Rehash** | 340.6 | 28.98 | 402.1 | 25.86 |
| AnimateDiff | Original | 758.7 | 28.89 | 607.1 | 29.40 |
| | DeepCache | 840.2 | 28.15 | 615.8 | 29.06 |
| | **Step Rehash** | 784.5 | 28.71 | 603.9 | 29.29 |

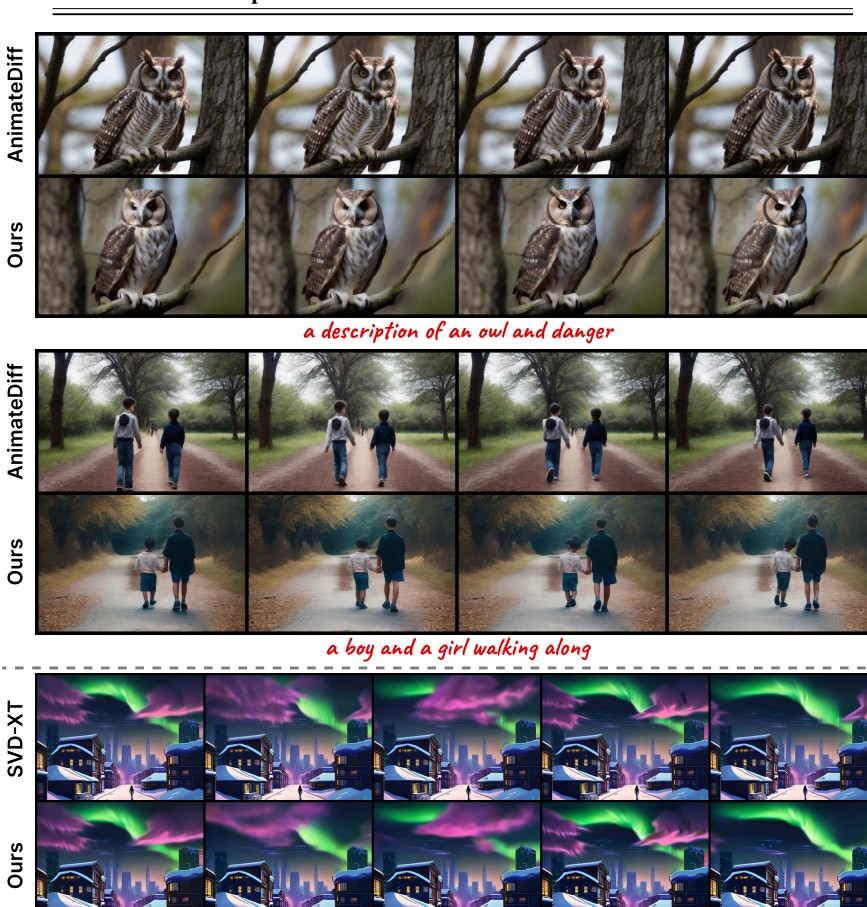

Figure 7: Quality evaluation of using our method on baseline models. The results show that our method can be generally applied to various video diffusion models and achieve competitive results.

## 5.4 Quality results

In Fig. 7 and Appendix E, we present qualitative results comparing our method to the original model without using Streamlined Inference.We can see that our method produces vivid and high-quality samples aligned with the text descriptions. More importantly, these results demonstrate that our method can significantly reduce peak memory and computation without sacrificing quality.

## 6 Conclusion and Limitation

In this paper, we propose a novel training-free framework that significantly reduces peak memory and computation costs for video diffusion model inference by leveraging its spatial and temporal characteristics. Our approach can be seamlessly integrated into existing models. Extensive experiments on

SVD, SVD-XT, and AnimateDiff demonstrate our method's effectiveness in reducing peak memory and accelerating inference without sacrificing quality. Our approach offers a new perspective for fast, memory-efficient video diffusion, enabling the generation of higher quality and longer videos on consumer-grade GPUs. Though our method is general, the efficiency is limited by baseline model architecture design.

## Acknowledgments and Disclosure of Funding

This work is supported by National Science Foundation CNS-2312158. We would like to express our sincere gratitude to the reviewers for their invaluable feedback and constructive comments to improve the paper.

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

# Appendix

## A Memory Snapshot during inference

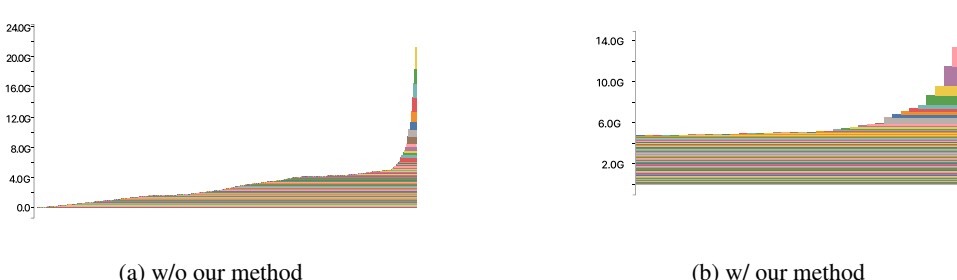

(a) w/o our method           (b) w/ our method

Figure A1: GPU memory snapshot of active cached segment timeline for Stable Video Diffusion with 14 frames $512 \times 512$

We provide memory snapshots under different configurations during inference, demonstrating the effectiveness of memory reduction. An example is shown in Fig.A1. This example shows the memory reduction of our method on SVD with $512 \times 512$ resolution. The snapshot is collected following the tutorial[2].

## B Key Step Search for Step Rehash

**Example of Step Rehash.** For step $s + 1$, we only conduct part of the computations in the final up-sampling block, skipping most of the computations in the U-net. Similarly, we can skip multiple steps. For example, if we skip both step $s + 1$ and $s + 2$, to obtain the output $U_3^{s+2}$ for step $s + 2$, we feed the output features $U_3^{s+1}$ into the final up-sampling block of step $s + 2$, where $U_3^{s+1}$ is also obtained from $U_3^s$ following the above reusing and skipping method.

**Similarity patterns.** The feature similarity between different steps exhibits certain patterns. As shown in Fig. 5a, at initial steps, the similarity is high (above 97%) across multiple steps such as from step 0 to step 13. In the middle steps, the high similarity only appears within a small step range. For example, the similarity between step 17 and step 19 is lower than 93%. In the final steps, the high similarity appears in a slightly larger step range, such as from step 20 to step 22, with above 93% similarity.

We address the where-to-skip problem with a fixed strategy to skip the computations from the specific blocks. Next, we address the when skip problem to choose what steps can be skipped based on the similarity map. Given the similarity map as shown in Fig. 5a, the similarity value between step $i$ and $j$ can be represented by $S_{ij}$ as shown in the similarity map. We develop a search method to find the key steps with feature rehash and skip the other steps.

The algorithm is shown in Algorithm 1. We use a threshold to select the key steps. If the similarity of multiple consecutive steps is above the threshold, we only select the start and end steps as key steps, and the middle steps can be skipped. Typically, a larger threshold leads to more key steps with high generation performance close to the original one, and a smaller threshold leads to skipping more steps and, thus, computations with faster generation.

---

**Algorithm 1** Key step search in step rehash

---

**Require:** The similarity map $\mathbf{S}$, the similarity threshold $\gamma$, the maximum step number $K$
**Ensure:** The set of key steps $G$
  $i \leftarrow 0, j \leftarrow 0, G \leftarrow \{\}, G \leftarrow G \| i$
  **while** $i < K$ **do**
    **if** $\mathbf{S}_{ij} \geq \gamma$ **then**
      $i \leftarrow i + 1$
    **else**
      $G \leftarrow G \| i$
      $j \leftarrow i$
  $G \leftarrow G \| K - 1$
  **return** $G$

---

[2]https://pytorch.org/blog/understanding-gpu-memory-1/

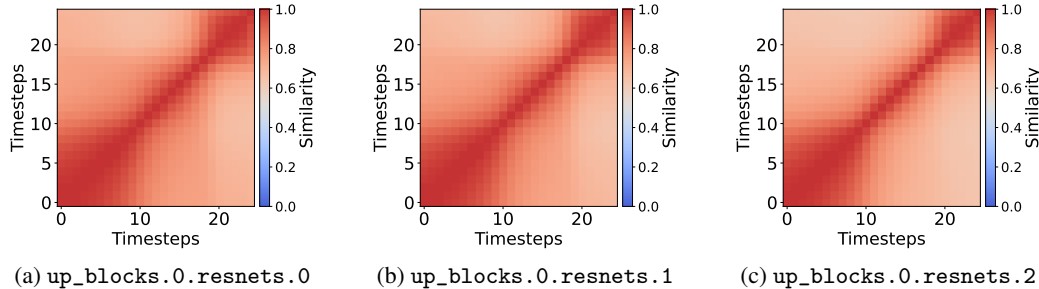

(a) `up_blocks.0.resnets.0`  (b) `up_blocks.0.resnets.1`  (c) `up_blocks.0.resnets.2`

Figure A2: Similarity maps of different temporal layers in `up_blocks.0.resnets`.

We provide sample PyTorch snippets for operation grouping and Step Rehash. The sample code effectively reduces the peak memory and accelerates the inference speed. However, the pipeline is not released because it requires specific compilation support.

## C  Similarity map of middle layers

We illustrate the similarity map of several layers closer to the *mid-block* of the UNet, showing that the similarity of these layers is relatively low compared to the results in Fig. 5.

## D  Visual comparison with DeepCache

We provide visual comparison of our method with DeepCache in here. As we can see, our method produces more vivid and detailed sample than DeepCache.

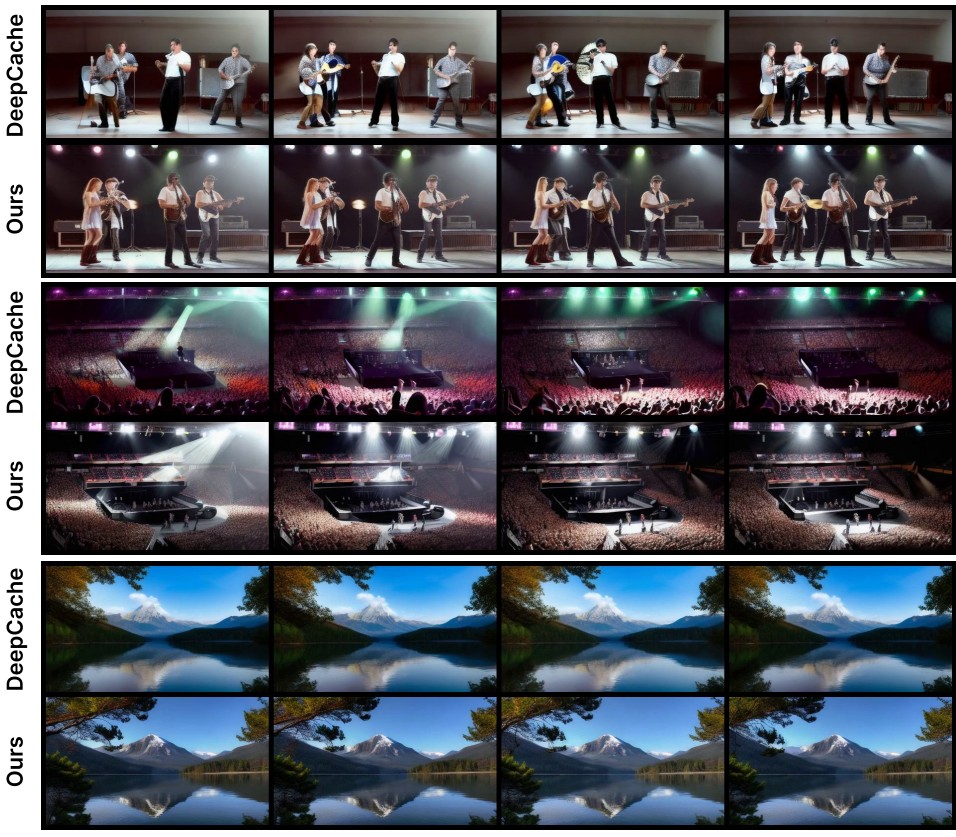

Figure A3: Visual comparison of our method with DeepCache.

# E   More quality results

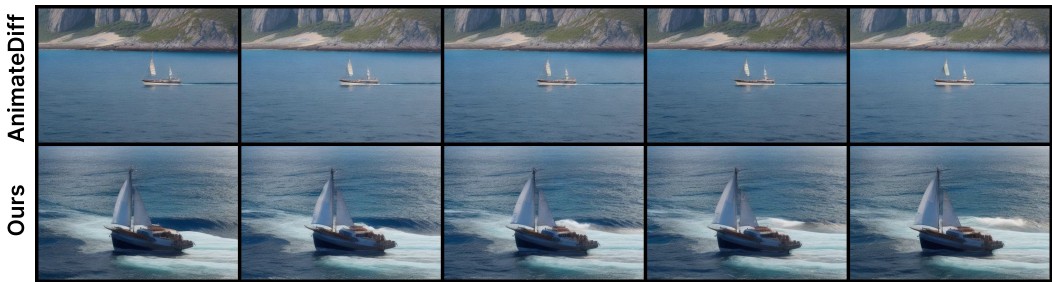

a boat sailing in the ocean with a rocky landmass in view

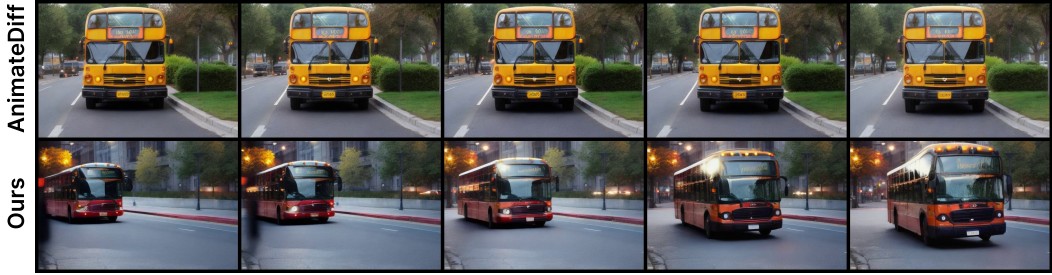

a bus pulls up to a curb then pulls off

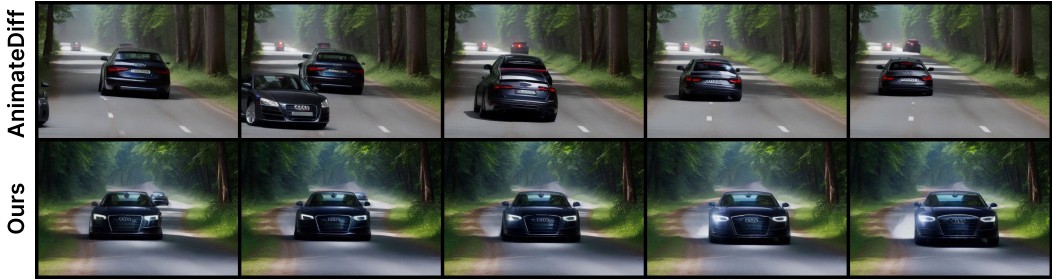

the expensive audi car going very fastly in
the road at the center of the forest

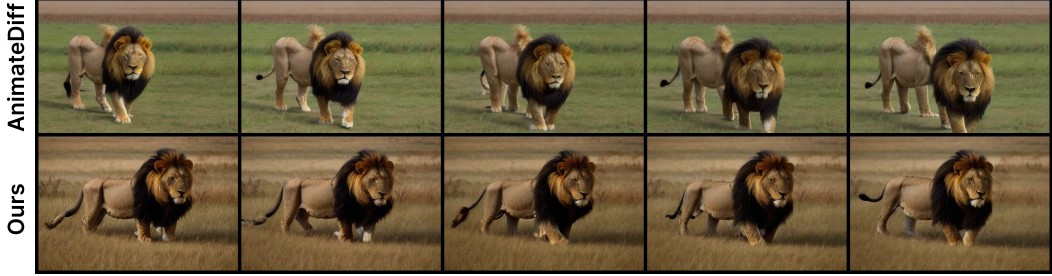

a lion is shown with another lion walking through the field

Figure A4: Quality evaluation of using our method on baseline models.

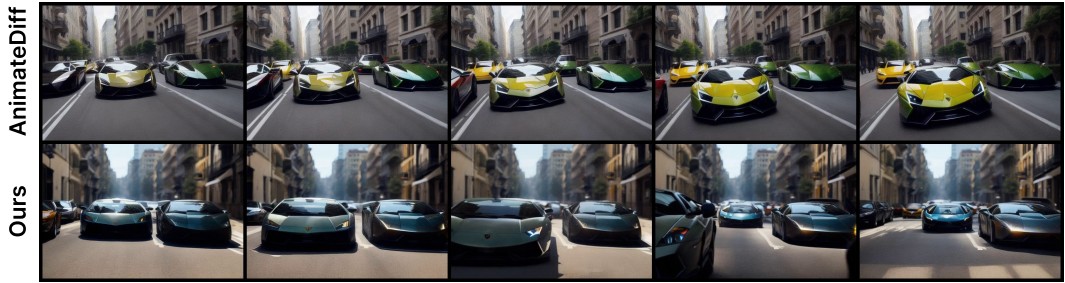

*a bunch of loud lamborghini s a driving up and down the street all at different times*

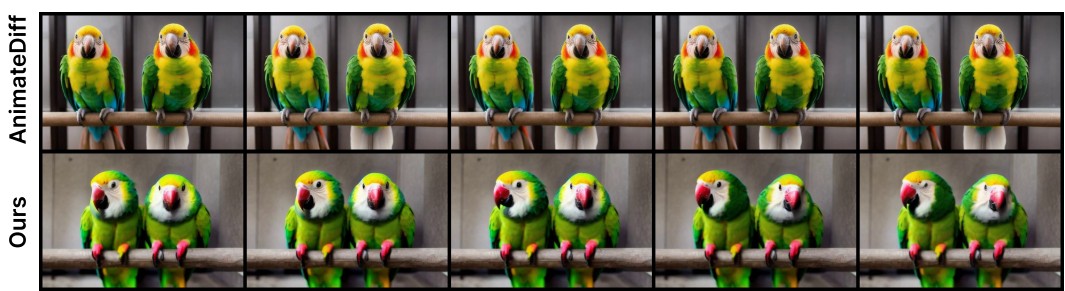

*2 cute parrots sitting nicely*

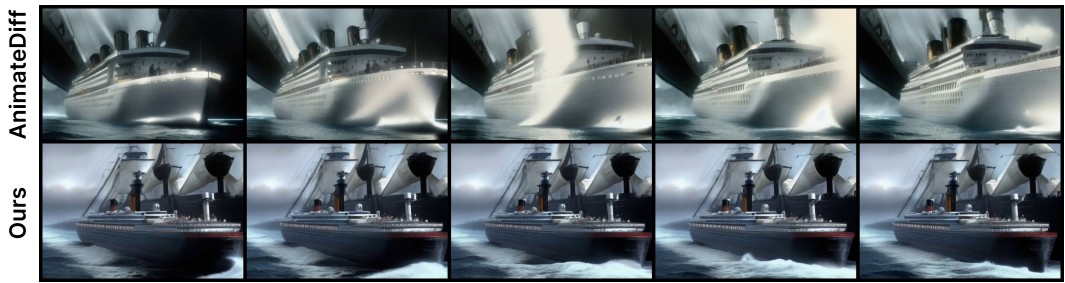

*a movie trailer of james cameron s titanic movie*

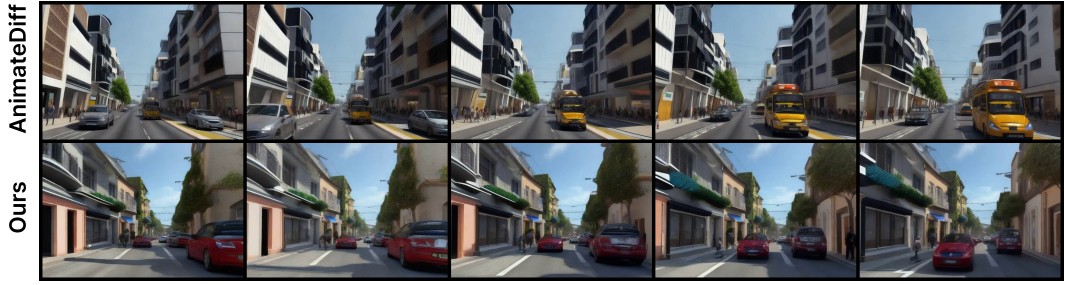

*a street with traffic shown first then a setting of
custom made convertables are being displayed*

Figure A5: Quality evaluation of using our method on baseline models.

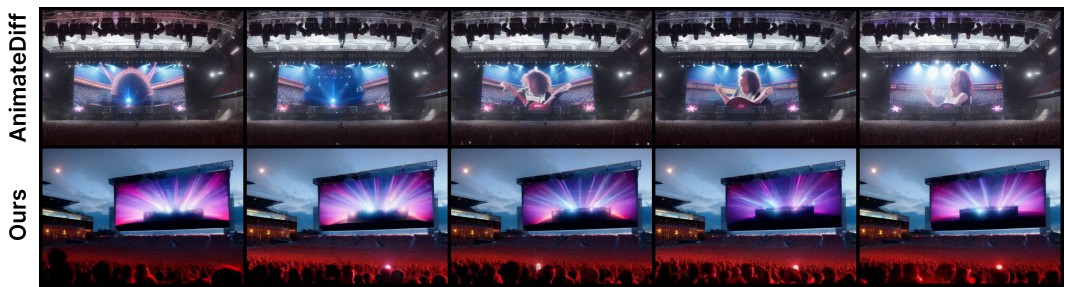

a band is performing and being shown on a large screen

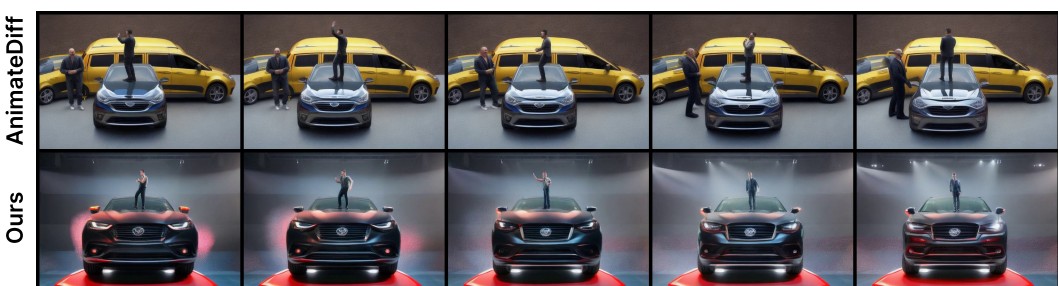

a man is standing on a car at a car tv show

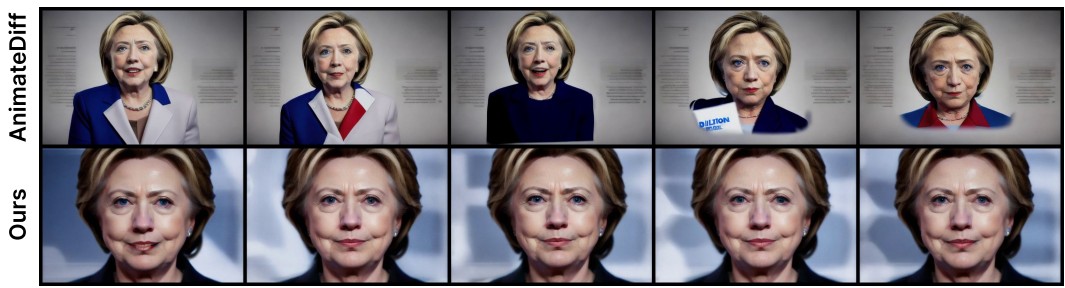

a campaign ad for hillary clinton

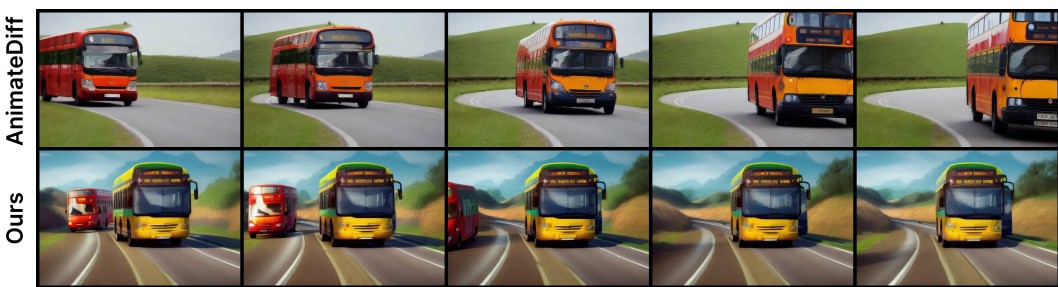

a beautiful rhyme about the roadbus journey for children

Figure A6: Quality evaluation of using our method on baseline models.

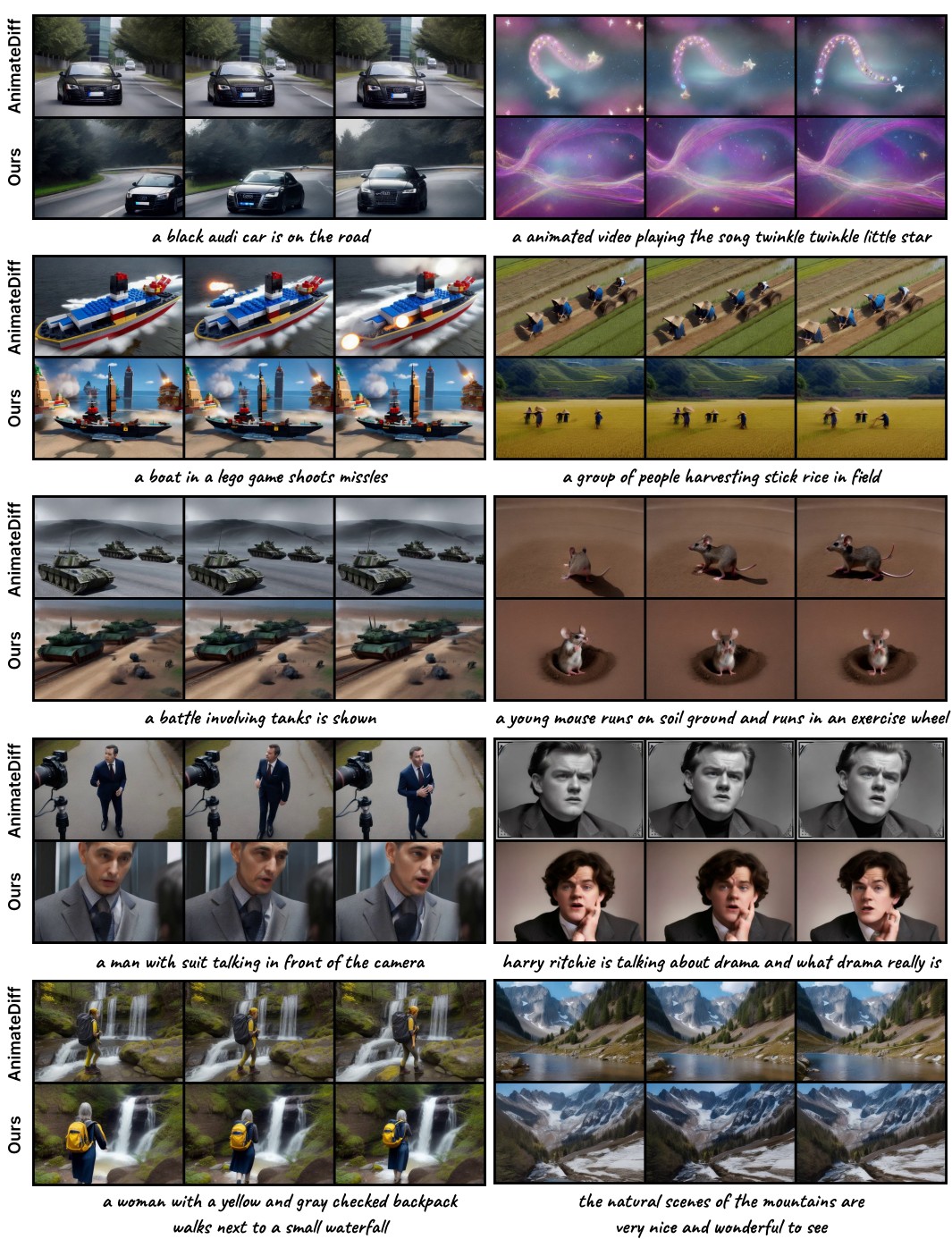

Figure A7: Quality evaluation of using our method on baseline models.

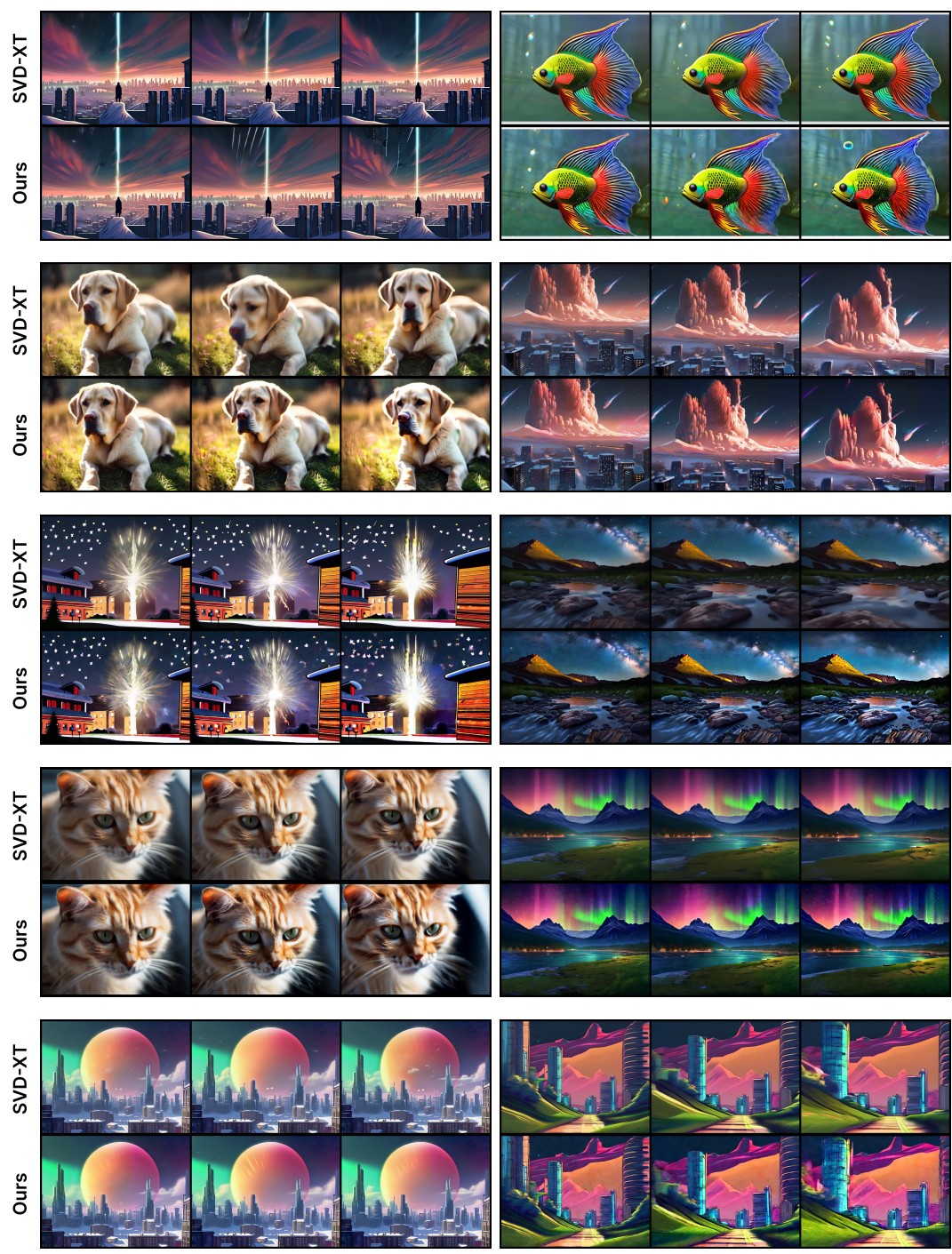

Figure A8: Quality evaluation of using our method on baseline models.

