# OpenReview forum: "Fast and Memory-Efficient Video Diffusion Using Streamlined Inference"
_NeurIPS.cc/2024/Conference — NeurIPS 2024 poster_

### Official Review · Reviewer_5EX5 · 2024-06-23

**Soundness:** 4
**Presentation:** 4
**Contribution:** 3
**Rating:** 6
**Confidence:** 5

**Summary:**

In this paper, the authors focus on reducing the computational requirements and high peak memory usage for video diffusion models. Specifically, a train-free framework is proposed, which consists of three parts: Feature Slicer, Operator Grouping, and Step Rehash. Those three steps result in significant memory reduction and inference acceleration.

**Strengths:**

1.	The proposed framework has a great performance for reducing peak memory and accelerating the inference of video diffusion models. In particular， the peak memory of AnimateDiff can be reduced significantly from 41.7GB to 11GB, which can contribute to more practical applications.

2.	The pipeline of the proposed framework is easy to understand and is well-documented.

**Weaknesses:**

1.	More base models with other frameworks of video diffusion should be involved for more comparisons. The two baselines used in experiments, i.e., SVD and AnimateDiff, are both based on T2I diffusion models and Unet backbone. Is the proposed framework suitable for other diffusion models with DiT backbone[1] or 3D diffusion, like Open-Sora Plan v1.1(can be found in Github).

2.	Human-level metrics should be involved to compare the video visual quality in Table 2.  The FVD and CLIP-Score may not be enough to measure the performance. Based on the cases shown in Figure 1 and 7, I prefer to agree that the original base model has better performance.


[1]: Peebles W, Xie S. Scalable diffusion models with transformers[C]//Proceedings of the IEEE/CVF International Conference on Computer Vision. 2023: 4195-4205.

**Questions:**

My main question is about the generalization of the proposed framework. Based on my understanding, most contributions of this paper are “engineering”. It is important to verify the performance on various base models, such as diffusion models with DiT backbone and 3D diffusion models.

In addition, if I want to use this framework, do I need to modify a lot of content in the code?

---

> ### Author Rebuttal · Authors · 2024-08-07
>
> We sincerely appreciate the reviewer for recognizing the strengths of our papers and providing valuable feedback. We are happy to address the raised questions as below.
>
> ---
> #### **W1.  Generalization on other variants and backbones.**
>
> We agree that a more comprehensive evaluation could help demonstrate the generalization ability of our framework. We evaluate DiT, ModelScope and VideoCrafter with our method, and demonstrate that our method is general and can be applied to multiple video diffusion models with different architectures.
> Please refer to Global **A3** Table 5~8 for more detailed results.
>
> ---
> #### **W2. More evaluation metrics and video quality.**
>
> We agree that it is important to provide a more comprehensive evaluation. Please refer to Global rebuttal **A1**, **A2** and **A4** for more detailed results. We include human evaluation in Table 4 in Global rebuttal **A2**, other evaluation metrics results can be found in Table 2 and Table 3 in Global rebuttal **A2**. As discussed in Global rebuttal **A1**, the video quality can be **maintained** by slightly increasing the full computation steps of Step Rehash while still **keeping a good amount inference acceleration**. In our paper, we  use **fewer full computation steps** to **push the limits** of Step Rehash. Our quantitative results in Table 1 of Global rebuttal **A1** and visual demonstrations in Global rebuttal **A4** demonstrate that our method can lead to high quality video generation with reduced cost.
>
> ---
> #### **Q1. Generalization of our method and contributions clarification.**
>
> First, we want to clarify that our work is trying to solve the memory bound issues of video diffusion due to users’ limited VRAM resources in a novel way. Previous training free works focusing on sampling efficiency without considering the huge peak memory requirement for diffusion models. General techniques like Model compression take substantial efforts to retrain or finetune the diffusion model to recover performance, which is costly, time-consuming, and may raise data privacy concerns. Furthermore, applying post-training compression techniques in one-shot may save the retraining/fine-tuning efforts, but suffers from significant performance degradation.
>
> Our work leverages the **system and algorithm co-design** (where the feature slicer + operator grouping can collaborate with step rehash without any interference) to provide a new direction for providing a fast and memory-efficient diffusion **without training**.
>
> Last but not least, we would like to kindly clarify that our method could be applied to DiT backbones without any design changes. Please refer to Global rebuttal **A3** Table 6 and Table 8 for more detailed results.
>
> ---
> #### **Q2. In addition, if I want to use this framework, do I need to modify a lot of content in the code?**
>
> We have implemented our method on all spatial-temporal 3D blocks in `diffusers` and will release a Python wrapper which can automatically inherit original blocks with our fast and memory-efficient blocks in various video diffusion models. Our implementation extends the original blocks and can seamlessly replace them without changing the diffusion framework. It is easy to implement.

---

> > ### Comment · Reviewer_5EX5 · 2024-08-13
> >
> > Thanks for your responses. I'm glad to improve my score.

---

> ### Author Response · Authors · 2024-08-12
>
> Dear Reviewer,
>
> Thank you very much for taking the time to review our paper and for acknowledging the performance of our framework. Since the discussion will end very soon, we sincerely hope that you have found time to check our detailed response to your previous questions/comments. If you have any further questions, please feel free to let us know. We will try our best to reply to you before the discussion deadline.
>
> Thank you very much,
>
> Authors

---

> ### Author Response · Authors · 2024-08-13
>
> We sincerely thank the reviewer for acknowledging our responses and improving the score! We will add all these constructive suggestions in the final version of our paper.

---

### Official Review · Reviewer_M5Qw · 2024-07-10

**Soundness:** 3
**Presentation:** 3
**Contribution:** 3
**Rating:** 6
**Confidence:** 2

**Summary:**

This paper proposes a training-free video diffusion inference acceleration method, which includes three processes: Feature Slicer, Operator Grouping, and Step Rehash. Compared to the baseline, the proposed method shows significant improvements in memory usage and inference speed.

**Strengths:**

- The method description is straightforward and easy to understand.
- The proposed method is effective, simple, and easy to implement, and it shows significant improvements.
- The problem addressed by the proposed method is critical.

**Weaknesses:**

- The experiments need to be improved. Step Rehash can greatly accelerate the inference speed, but it seems to be highly dependent on the weights of the video diffusion model. It is unclear whether the proposed method can be applied to most video diffusion models. Therefore, the authors need to compare more types of models and provide statistical information on the performance changes, such as mean and variance.
- The proposed method shows a visible loss in video generation capability, as seen in Figure 7 where the owl's eyes disappear and the overall texture quality of the synthesized image decreases in the optimized version.

**Questions:**

- From the images, it is clear that the quality of some of the synthesized images has decreased. Based on my experience, this decrease in quality may have a more severe impact on video synthesis tasks. The authors did not seem to provide video demos, so it is difficult to judge how severe the loss in performance is.

- What is the relationship between the optimization method and batch size used during inference, and how do they affect the optimization speed and memory usage?

**Limitations:**

I think this paper addresses an important problem and appears to have significant improvements. However, the degree of loss in synthesis quality is not clear, which is an area of concern for me. The loss in quality seems to be less noticeable in SVD, but more significant in AnimateDiff (e.g., the owl in Figure 7, the bus in Figure A6, and the parrot in Figure A5).

---

> ### Author Rebuttal · Authors · 2024-08-07
>
> We sincerely appreciate the reviewer for recognizing the strengths of our papers and providing valuable feedback. We are happy to address the raised questions as below.
>
> ---
> #### **W1. The experiments need to be improved.**
>
> Please refer to Global rebuttal **A3** and **A2** for more detailed results.
>
> We agree that a more comprehensive evaluation could help demonstrate the generalization ability of our framework. Therefore, we conduct comprehensive experiments on various model architectures (Global rebuttal **A3**) and demonstrate more evaluation metrics results with statistical information (Global rebuttal **A2**). Our extended experiment results show that our framework could generalize to different backbones/architectures and significantly reduces peak memory and computation costs for video diffusion model inference.
> We would also like to kindly point out that we already provide the mean of FVD, and FVD results do not support variance evaluation.
>
> ---
> #### **W2. Concern about maintaining video quality.**
>
> Please refer to Global rebuttal **A1** and **A4**  for more detailed results. As discussed in Global rebuttal **A1**, the video quality can be **maintained** by slightly increasing the full computation steps of Step Rehash while still **keeping a good amount inference acceleration**. In our paper,  we  use **fewer full computation steps** to **push the limits** of Step Rehash. Our quantitative results in Table 1 of Global rebuttal **A1** and visual demonstrations in Global rebuttal **A4** demonstrate that our method can lead to high quality video generation with reduced cost.
>
> ---
> #### **Q1. Concern about maintaining video quality.**
>
> Please refer to Global rebuttal **A1** and **A4**.
>
> ---
> #### **Q2. Relationship between the optimization method and batch size.**
>
> Thank the reviewer for raising this valuable question. We implement this part and find our method could handle the scalability in terms of batch size.
> Our slicing strategy will be accordingly adjusted based on the batch size and input size.
> Below is the performance benchmark on batch size = 1,2,4 for AnimateDiff with 512x512 output.
> - Table A, when batch size = 1,2,4
> | **Model** | **Batch Size** | **Peak Mem** | **Latency** |
> |:-:|:-:|:-:|:-:|
> | AnimateDiff+Ours | 1 | 7.51G | 7.08s |
> | AnimateDiff+Ours | 2 | 8.30G | 14.07s |
> | AnimateDiff+Ours | 4 | 11.35G | 27.05s |
>
> ---
> #### **L1. Concern about maintaining video quality.**
>
> Please refer to Global rebuttal **A1** and **A4**.

---

> ### Author Response · Authors · 2024-08-12
>
> Dear Reviewer,
>
> Thank you very much for taking the time to review our paper and for acknowledging the contributions we've made. Since the discussion will end very soon, we sincerely hope that you have found time to check our detailed response to your previous questions/comments. If you have any further questions, please feel free to let us know. We will try our best to reply to you before the discussion deadline.
>
> Thank you very much,
>
> Authors

---

> > ### Comment · Reviewer_M5Qw · 2024-08-13
> >
> > Thanks for your responses, which has addressed some of my concerns. I will raise my score.

---

> > > ### Author Response · Authors · 2024-08-13
> > >
> > > We sincerely thank the reviewer for recognizing that the concerns have been addressed and raising the score! We will add all these constructive suggestions in the final version of our paper.

---

### Official Review · Reviewer_2uzE · 2024-07-12

**Soundness:** 2
**Presentation:** 3
**Contribution:** 3
**Rating:** 6
**Confidence:** 4

**Summary:**

This paper presents a framework for reducing the computational demands of text-to-video diffusion models. The main idea involves dividing input features into subfeatures and processing them parallelly, thereby reducing peak memory usage during sampling. To address the increase in overall compute time caused by this partitioning, the authors propose a skip strategy that determines when and where to apply the skip operation based on feature similarities.

**Strengths:**

1. The paper is well-written and motivated.
2. The proposed method is intuitive.

**Weaknesses:**

While the paper is well-motivated for the important problem and proposes an intuitive and simple remedy, there are major concerns regarding the evaluation, scalability, and applicability of the proposed method, as follows:

**Scalability and Applicability**
- My major concern is the scalability of the proposed method. Since the method requires dividing the input features into patches, it inevitably increases the computation as the feature dimension gets larger, potentially introducing more patches to compute. While the proposed skip strategy might mitigate some of the increased computation time, it definitely harms the quality of the generated videos.

- In terms of applicability, the proposed framework is only applicable to U-Net-based video diffusion models. However, recent models, such as those utilizing DiT backbones, are not addressed by this framework.

**The evaluation is weak**
- Lack of baselines: The authors only compare their method to naive hashing, which is the most simple baseline. To verify the effectiveness of the proposed method, please include existing method for efficient sampling that also reduce memory and sampling speed. The authors only compare with DeepCache. They should compare with more training-free baselines, even though these primarily demonstrate their efficiency in image diffusion models. If not applicable, discuss why extending these approaches to the text-to-video diffusion model is not straightforward in the Related Work section, and also discuss the differences with existing baselines.
- Week evaluation with DeepCache: It is unclear why the authors made a comparison with DeepCache under the same computational step. The authors should define what they mean by “computational step” and compare with DeepCache in terms of efficiency metrics as thoes in Table 1. Also, the metric of MAC and GFlops should be included to measure efficiency.
- Evaluation protocol: The current evaluation using FVD and CLIP score is insufficient. It would be beneficial to include more comprehensive video quality evaluation metrics, such as thoes proposed in VBench [1].
- Additional experiments: Include more video diffusion backbones such ModelScope and VideoCrafter to extensively verify the effectiveness of proposed method under various video diffusion models.

[1] Huang et al., VBench: Comprehensive Benchmark Suite for Video Generative Models, CVPR 2024

**Questions:**

What is the "computational step" in Table 2?

**Limitations:**

See the Weaknesses Section.

---

> ### Author Rebuttal · Authors · 2024-08-07
>
> We sincerely appreciate the feedback from the reviewer. First, we would like to kindly clarify some misunderstandings here:
> 1. We do provide the definition of the “full computation step” in the “line 265, 266”.
> 2. Our method can be applied to DiT backbones without any design changes.
> 3. Our work takes an early step in fast and memory-efficient inference framework in a training free manner.
>
> We address the raised questions as below.
>
> ---
> **W1. Scalability concern.**
>
> We would like to point out that our observation is that our method could actually mitigate I/O overhead which can handle the scalability issues, as stated in “Section 4.2.1 **Mitigating I/O intensity**”. Moreover, we introduce the pipeline to mitigate the “more patches” things as stated in “Section 4.2.2”. Overall, our experimental results use 3x576x1024 with up to 16 frames which is considered as large scale video generation for the current open-source SOTA video diffusion models.
>
> We provide a detailed breakdown, as we can see, even at 576x1024 with 16 frames, our method without step rehash could still mitigate the overhead from the scale. And we use full computation steps=15 for maintaining the quality of the generated videos. Here are the results.
> - Table A, proposed method breakdown
> |Model|Method|speed up|Peak Mem. (576x1024)|FVD (UCF101)|CLIP-Score (UCF101)|
> |-|:-:|:-:|:-:|:-:|:-:|
> |SVD|-|-|39.49G|307.7|29.25|
> ||+ Feature Slicer|x0.95|39.49G|307.7|29.25|
> ||+ Feature Slicer + Operator Grouping|x0.98|23.42G (-40.7%)|307.7|29.25|
> ||+ Feature Slicer + Operator Grouping + Pipeline|x1.03|23.42G (-40.7%)|307.7|29.25|
> ||+ Feature Slicer + Operator Grouping + Pipeline + Step Rehash|x1.46|23.42G (-40.7%)|312.1|29.20|
> |AnimateDiff|-|-|41.71G|758.7|28.89|
> ||+ Feature Slicer|x0.94|41.71G|758.7|28.89|
> ||+ Feature Slicer + Operator Grouping|x0.96|11.07G (-73.5%) |758.7|28.89|
> ||+ Feature Slicer + Operator Grouping + Pipeline|x1.03|11.07G (-73.5%)|758.7|28.89|
> ||+ Feature Slicer + Operator Grouping + Pipeline + Step Rehash|x1.45|11.07G (-73.5%)|765.01|28.87|
>
> We also provide results for scalability of batch_size to further demonstrate our method. Here are the scalability results of our method when batch size is not 1. Our slicing strategy will be accordingly adjusted based on the batch size and input size. Below is the performance benchmark on batch size = 1,2,4 for AnimateDiff with 512x512 output. We can see that when the batch_size becomes 4, the peak memory does not increase significantly compared with the batch_size of 1.
> - Table B, scalability under batch size = 1,2,4
> |Model|Batch Size|Peak Mem|Latency|
> |-|:-:|-|-|
> |AnimateDiff+Ours|1|7.51G|7.08s|
> |AnimateDiff+Ours|2|8.30G|14.07s|
> |AnimateDiff+Ours|4|11.35G|27.05s|
>
> ---
> **W2. Generalization to DiT backbones.**
>
> Please refer to Global rebuttal **A3** for more detailed results.
>
> We would like to kindly clarify that our method could be applied to DiT backbones without any design changes.
> This is due to our general design of Feature slicer on spatial-temporal 3D blocks in video diffusion blocks. We would also like to point out that Operator Grouping, Pipeline and Step Rehash are general techniques. Our results on DiT backbones (OpenSora) demonstrates stable generalization on various backbones/architectures and can constantly improving efficiency while maintaining the video quality.
>
> ---
> **W3. Lack of baselines.**
>
> We would like to kindly clarify some misunderstandings here.  Our work tries to solve the memory bound issues of video diffusion due to users’ limited VRAM resources in a novel way. Current sampling methods focus on improving sampling efficiency, without any considerations of reducing the huge peak memory where "Peak Memory" is the **maximum** amount of memory which is used by the processes during the **entire time**. Also, DeepCache is the most recent training-free baselines that introduce a novel direction for improving inference speed **only** in image diffusion models. General techniques like Model compression for fast and memory-efficient models take substantial efforts to retrain or finetune the diffusion model to recover performance. To our best knowledge, our work takes an early step in fast and memory-efficient inference framework in a training free manner.
>
> We survey several methods that can reduce diffusion inference step such as DPM-Solver and Euler solver. However, they can not reduce the peak memory during inference which we highlight as one of the main contributions of our method.
>
> We would be thankful if the reviewer could give more details of “lacked baselines” about training-free baselines for memory efficiency in diffusion models so we could include more comprehensive experimental results.
>
> ---
> #### **W4. Week evaluation with DeepCache.**
>
> We would first like to clarify that we provide the definition of the “full computation step”  in the “line 265, 266” for steps that can't be skipped; other steps that use step rehash have most of their computation skipped as we stated in Section 4.3.2. Moreover, we would like to point out that our method can save more GFLOPS when under the same computational step as stated in “line 321, 322”. We provide detailed results below,
> - Table C, Computation Comparison
> |Model|Method|512x512|576x1024|
> |-|-|-|-|
> |SVD|Original|8.47T|20.40T|
> ||DeepCache|5.68T|14.06T|
> ||Ours|5.75T|13.25T|
> |AnimateDiff|Original|8.37T| 20.19T|
> ||DeepCache|5.76T|14.09T|
> ||Ours|5.53T|11.91T|
>
> ---
> **W5. Evaluation protocol.**
>
> Please refer to Global rebuttal **A2**, Table 2~4.
>
> ---
> **W6. Additional experiments.**
>
> Please refer to Global rebuttal **A3**, Table 5 and Table 7.
>
> ---
> **Q1. What is the "computational step" in Table 2?**
>
> We would like to point out that we provide the definition of the “full computation step” in the “line 265, 266” for steps that can't be skipped; other steps that use step rehash have most of their computation skipped as we stated in Section 4.3.2.

---

> > ### Comment · Reviewer_2uzE · 2024-08-13
> >
> > Thank you very much for the detailed responses. My concerns have been thoroughly addressed, and I have accordingly increased my score.

---

> > > ### Author Response · Authors · 2024-08-13
> > >
> > > We sincerely thank the reviewer for recognizing that the concerns have been thoroughly addressed and increasing the score! We will add all these constructive suggestions in the final version of our paper.

---

> ### Author Response · Authors · 2024-08-12
>
> Dear Reviewer,
>
> Thank you very much for spending time reviewing our paper. Since the discussion will end very soon, we sincerely hope that you have found time to check our detailed response to your previous questions/comments. If you have any further questions, please feel free to let us know. We will try our best to reply to you before the discussion deadline.
>
> Thank you very much,
>
> Authors

---

> ### Comment · Area_Chair_RSqD · 2024-08-13
>
> Dear Reviewer,
>
> As we enter the final day of the Reviewer-Author discussion, please take a moment to review the authors' rebuttal and consider the comments from other reviewers. If you have any additional questions, now is the time to ask them to help you better evaluate the paper. Thanks!

---

### Official Review · Reviewer_mL5N · 2024-07-14

**Soundness:** 3
**Presentation:** 3
**Contribution:** 3
**Rating:** 6
**Confidence:** 3

**Summary:**

The paper introduces a novel, training-free framework to optimize video diffusion models. This framework, consisting of Feature Slicer, Operator Grouping, and Step Rehash, significantly reduces peak memory usage and computational overhead while maintaining video quality. Extensive experiments demonstrate that the approach can cut memory usage by up to 70% and improve inference speed by 1.6 times compared to baseline methods. The framework is compatible with existing models like AnimateDiff and SVD, enabling high-quality video generation on consumer-grade GPUs. The research paves the way for more efficient video diffusion models, making advanced video generation accessible on standard hardware.

**Strengths:**

- The introduction of a training-free framework that optimizes video diffusion models seems a novel approach.
- The method is compatible with existing video diffusion models like AnimateDiff and SVD, ensuring broad applicability.
- Comprehensive experiments and detailed analysis demonstrate the framework's effectiveness and robustness.

**Weaknesses:**

- Although the paper claims that video quality is maintained, the extent of quality degradation, if any, is not fully quantified.
- Although some ablation studies are provided, a more detailed breakdown of the contributions of each component (Feature Slicer, Operator Grouping, and Step Rehash) would strengthen the understanding of their individual and combined impacts.
- The evaluation relies heavily on FVD and CLIP-Scores, which, while useful, may not capture all dimensions of video quality and user satisfaction. Including additional metrics or user studies could provide a more holistic assessment of the generated video quality.

**Questions:**

- Are there any specific scenarios where the overhead introduced by slicing and grouping operations could outweigh the benefits?
- Have you considered conducting user studies to provide a more comprehensive assessment of the generated video quality?

**Limitations:**

Authors have adequately addressed the limitations.

---

> ### Author Rebuttal · Authors · 2024-08-07
>
> We sincerely appreciate the reviewer for recognizing the strengths of our papers and providing valuable feedback. We are happy to address the raised questions as below.
>
> ---
> #### **W1. Concern about maintaining video quality.**
>
> Please refer to Global rebuttal **A1** and **A4**. As discussed in Global rebuttal **A1**, the video quality can be **maintained** by slightly increasing the full computation steps of Step Rehash while still **keeping a good amount inference acceleration**. In our paper, we  use **fewer full computation steps** to **push the limits** of Step Rehash. Our quantitative results in Table 1 of Global rebuttal **A1** and visual demonstrations in Global rebuttal **A4** demonstrate that our method can lead to high quality video generation with reduced cost.
>
> ---
> #### **W2. Need more breakdown to our method.**
>
> Thank the reviewer for pointing out this part. We will provide more details and rephrase it for better clarity. We also provide more breakdown results for this question. First, we would like to kindly point out that:
>
> As mentioned in “Line 165-167”, Feature Slicer cannot reduce peak memory if not combined with Operator Grouping.
>
> The Operator Grouping can only be applied after slicing the features. It is a solution to reduce saving intermediate results after applying the Feature Slicer. We would also like to point out that our Operator Grouping, Pipeline and Step Rehash are general techniques.
>
> From the following table, we can see that Feature Slicer and Operator Grouping together can significantly reduce the peak memory.  Pipeline and Step Rehash lead to further accelerations. Step Rehash incurs more significant speedups with certain generation quality degradation. Pipeline does not affect the video quality.
>
> ---
> - Table 1. Breakdown of our method (the results here are aligned with results Table 1 from our paper)
> | **Model** | **Method** | **speed up** | **Peak Mem.  (576x1024)** | **FVD (UCF101)** | **CLIP-Score (UCF101)** |
> |:---:|:---:|:---:|:---:|:---:|:---:|
> | SVD | - | - | 39.49G | 307.7 | 29.25 |
> |  | + Feature Slicer | x0.95 | 39.49G | 307.7 | 29.25 |
> |  | + Feature Slicer + Operator Grouping | x0.98 | 23.42G (-40.7%) | 307.7 | 29.25 |
> |  | + Feature Slicer + Operator Grouping + Pipeline | x1.03 | 23.42G (-40.7%) | 307.7 | 29.25 |
> |  | + Feature Slicer + Operator Grouping + Pipeline + Step Rehash | x1.63 | 23.42G (-40.7%) | 340.6 | 28.98 |
> | AnimateDiff | - | - | 41.71G | 758.7 | 28.89 |
> |  | + Feature Slicer | x0.94 | 41.71G | 758.7 | 28.89 |
> |  | + Feature Slicer + Operator Grouping | x0.96 | 11.07G (-73.5%) | 758.7 | 28.89 |
> |  | + Feature Slicer + Operator Grouping + Pipeline | x1.03 | 11.07G (-73.5%) | 758.7 | 28.89 |
> |  | + Feature Slicer + Operator Grouping + Pipeline + Step Rehash | x1.61 | 11.07G (-73.5%) | 784.5 | 28.71 |
>
> ---
> #### **W3. Need more evaluation metrics.**
>
> We agree that it is important to provide a more comprehensive evaluation. Please refer to Global rebuttal **A2**, Table 2 and Table 3. We demonstrate more evaluation metrics such as PSNR, LPIPS, SSIM, IS and Vbench benchmark. Our method can outperform the DeepCache baseline under various metrics.
>
> ---
> #### **Q1. Specific scenarios where the overhead introduced by slicing and grouping operations could outweigh the benefits.**
>
> When the generated video size is small enough like 64x64, the acceleration and memory reduction is not very obvious. However, in this case, even without any efficiency optimization methods, a consumer GPU is sufficient enough to handle this small size without difficulties.
>
> ---
> #### **Q2. User studies for more comprehensive assessment.**
>
> We agree that it is important to provide a more comprehensive evaluation. Please refer to Global rebuttal **A2**, Table 4.

---

> ### Author Response · Authors · 2024-08-12
>
> Dear Reviewer,
>
> Thank you very much for taking the time to review our paper and for acknowledging the novelty and applicability of our work. Since the discussion will end very soon, we sincerely hope that you have found time to check our detailed response to your previous questions/comments. If you have any further questions, please feel free to let us know. We will try our best to reply to you before the discussion deadline.
>
> Thank you very much,
>
> Authors

---

### Author Rebuttal · Authors · 2024-08-07

We thank the reviewers for acknowledging that our work importance and broad applicability (Reviewer mL5N, M5Qw), our method is novel, and high-performing (Reviewer mL5N, M5Qw, 5EX5), our experiments are comprehensive (Reviewer mL5N), and our paper is well-written (Reviewer 2uzE, 5EX5, M5Qw).

---
**A1. Concern about maintaining video quality.**

We would like to clarify that video quality can be **maintained** by slightly increasing the full computation steps of Step Rehash while still **keeping a good amount inference acceleration**. In our paper, the intention of our demonstrated results is using **fewer full computation steps** to **push the limits** of Step Rehash. To better illustrate our method's ability to maintain video quality, we provide additional results below and in **A4**.
- Table1. Quantitative results comparison under full computation step=15
|Model|Method|Full computation step|Speed Up|Peak Mem.(576x1024)|ucf101 FVD↓|ucf101 CLIP-score↑|MSR-VTT FVD↓|MSR-VTT CLIP-score↑|
|-|:-:|:-:|:-:|:-:|:-:|:-:|:-:|:-:|
|SVD|-|25|-|39.49G|307.7|29.25|373.6|26.06|
||+DeepCache|15|x1.39|39.49G|385.4|28.89|412.4|25.73|
||+Ours|15|x1.46|23.42G(-40.7%)|312.1|29.20|382.8|25.99|
|AnimateDiff|-|25|-|41.71G|758.74|28.89|607.1|29.40|
||+DeepCache|15|x1.39|41.71G|810.93|28.72|608.2|29.16|
||+Ours|15|x1.45|11.07G(-73.5%)|765.01|28.87|599.1|29.39|

Compared to the baseline, our Step Rehash outperforms DeepCache in both quality and speed. Specifically, when using the same full computation steps, Step Rehash skips more computations than DeepCache with less performance drop.

Lastly, we highlight that our work is an **early step** towards a **fast** and **memory-efficient** inference framework in a **training-free** manner. It reduces peak memory and computational overhead, making it feasible to generate high-quality videos on a single consumer GPU (e.g., reducing peak memory of AnimateDiff from 42GB to 11GB, with faster inference on a 2080Ti).

---
**A2. More evaluation metrics.**

We finish evaluating PSNR, LPIPS, SSIM, IS and Vbench benchmark with our method under full computation step=13. Our framework demonstrates stable generalization ability and can constantly maintain the video quality across various evaluation metrics according to results in Table 2~4. Here are the results,
- Table 2. More metrics
|Dataset|Method|PSNR↑|LPIPS↓|SSIM↑|IS↑|
|:-:|:-:|-|-|-|-|
|ucf101|SVD+Ours|22.44±3.60|0.27±0.08|0.72±0.12|27.11±0.54|
||SVD+DeepCache|17.07±3.14|0.41±0.09|0.52±0.16|25.69±0.33|
||AnimateDiff+Ours|16.94±2.28|0.54±0.08|0.51±0.09|27.89±0.68|
||AnimateDiff+DeepCache|16.31±3.14|0.54±0.12|0.50±0.12|27.62±0.47|
|msr_vtt|SVD+Ours|25.13±5.39|0.23±0.08|0.80±0.10|21.65±0.23|
||SVD+DeepCache|22.66±4.23|0.29±0.08|0.73±0.13|21.26±0.42|
||AnimateDiff+Ours|15.90±2.24|0.53±0.08|0.48±0.12|28.83±0.51|
||AnimateDiff+DeepCache|11.71±1.72|0.72±0.06|0.31±0.11|25.60±0.72|
- Table 3. Vbench Benchmark
|Dataset|Model|Method|Subject Consistency↑|Aesthetic Quality↑|Dynamic Degree↑|
|-|-|-|:-:|:-:|:-:|
|ucf101|SVD|-|0.92|0.39|0.45|
|||+Ours|0.89|0.36|0.33|
|||+DeepCache|0.88|0.34|0.62|
||AnimateDiff|-|0.94|0.50|0.77|
|||+Ours|0.93|0.49|0.80|
|||+DeepCache|0.92|0.49|0.80|
|msr_vtt|SVD|-|0.93|0.42|0.69|
|||+Ours|0.91|0.40|0.65|
|||+DeepCache|0.89|0.39|0.64|
||AnimateDiff|-|0.95|0.53|0.68|
|||+Ours|0.94|0.51|0.62|27.64±0.59|
|||+DeepCache|0.93|0.51|0.66|

We report human evaluation results in Table 4. via MTurk platform,
- Table 4. Human Preference Evaluation
|Method|Original (Win Rate)|Ours (Win Rate)|
|-|-|-|
|SVD|50.0%|50.0%|
|AnimateDiff|53.3%|46.7%|
|OpenSora|53.3%|46.7%|

---
**A3. Generalization on other variants and backbones.**

To demonstrate the generalization of our method, we evaluate DiT, ModelScope and VideoCrafter with our method. Our method is **general** and can be applied to other video diffusion variants. It **does not need any design change** for **adapting** to these **new** video diffusions. Specifically, we would like to point out that the generalization of Feature Slicer is achieved as all video diffusion models have temporal and spatial blocks for handling temporal and spatial information. Moreover, our Operator Grouping, Pipeline and Step Rehash **do not depend on the model architecture** which are **general techniques**. Our results in Table 5~8 demonstrates the stable generalization of our method on various backbones/architectures, and it can constantly improve efficiency while maintaining the video quality.
- Table 5. Quality on ModelScope and VideoCrafter2
|Dataset|Model|FVD↓|CLIP-score↑|
|-|:-:|-|:-:|
|ucf101|ModelScope|842.42|28.97|
||ModelScope+Ours|875.67|28.45|
||VideoCrafter2|823.11|29.02|
||VideoCrafter2+Ours|852.09|28.56|
|msr_vtt|ModelScope|846.90|25.65|
||ModelScope+Ours|868.01|25.14|
||VideoCrafter2|778.32|26.04|
||VideoCrafter2+Ours|810.66|25.54|
- Table 6. Quality on OpenSora
|Method|Subject Consistency↑|Aesthetic Quality↑|Dynamic Degree↑|
|-|:-:|:-:|:-:|
|OpenSora|94.45%|56.18%|47.22%|
|OpenSora+Ours|93.01%|54.82%|51.29%|
- Table 7. Efficiency results on ModelScope and VideoCrafter2.
|Model|Peak Mem.(576x1024)|Latency|
|-|:-:|-|
|ModelScope|12.51G|27.17s|
|ModelScope+Ours|8.30G|18.96s|
|VideoCrafter|Error*|Error*|
|VideoCrafter+Ours|14.16G|141.34s|

[*] Error occurred because VideoCrafter uses Xformer’s `memory_efficient_attention` which does not support the large memory scale when video size=576x1024. We replace the Attention with PyTorch `scaled_dot_product_attention`, the peak mem is 66.66G, and corresponding latency is 229.00s.
- Table 8, Efficiency results on OpenSora
|Model|Peak Mem (720P, 1:1)|Latency|
|-|:-:|-|
|OpenSora|59.91G|1230s|
|OpenSora+Ours|41.65G|952s|

---
**A4. Visualization results of videos.**

Due to the restriction of anonymous link, we provide a PDF contain every frame of generated videos (totally 30 videos with 15 video pairs) with our method on base models to better demonstrate that our method can lead to high quality video generation with reduced cost.

---

### Decision · Program_Chairs · 2024-09-25

**Decision:**

Accept (poster)

**Comment:**

The paper introduces a novel, training-free framework aimed at reducing the computational and memory demands of video diffusion models. The approach shows substantial improvements in memory efficiency (up to 70% reduction) and inference speed (1.6x faster) when compared to existing methods. This is particularly valuable for enabling high-quality video generation on consumer-grade hardware.

Initially, reviewers had concerns regarding the scalability and applicability of the proposed method, as well as the comprehensiveness of the evaluation settings. However, the authors effectively addressed these concerns during the rebuttal, providing extensive additional experimental evidence. This led to a consensus among all reviewers, who ultimately rated the paper as a weak accept.

Given the innovation, the thoroughness of the experimental validation, and the successful rebuttal addressing the initial concerns, I recommend accepting this paper.